# Effects of Graphene Morphology on Properties of Carbon Nanotube/Polyurethane Film Strain Sensors

**Chen Liu [1], Xiang Ge [2], Jiaqi Geng [1], Yuanli Men [1] and Caideng Yuan [1,*]**

[1] School of Chemical Engineering and Technology, Tianjin University, Tianjin 300350, China
[2] School of Mechanical Engineering, Tianjin University, Tianjin 300350, China
* Correspondence: cdyuan@tju.edu.cn

**Abstract:** The film flexible sensors can convert tiny changes in size or force into electrical signals. They are key components of intelligent devices and wearable devices, and are widely used in human-computer interaction, electronic skin, health monitoring, implantable diagnosis, and other fields. This kind of sensor is generally composed of polymer matrix and conductive components, while carbon nanotubes (CNT) and graphene (GN), as typical one-dimensional and two-dimensional conductive carbon nano-materials, respectively, have been used to build film flexible sensors. In order to explore the relationship between the GN size and thickness, and the performance of film sensors, the GN-CNT/PU composite film sensors were prepared by in situ polymerization of polyurethane (PU) in the presence of GN and CNT. A highly sensitive GN-CNT/PU flexible film sensor was prepared with a high gauge factor (GF) up to 13.15 in a strain range of 0–20%; an exceptionally low percolation threshold of GN is about 0.04 vol% when the CNT content is fixed at 0.2 vol%, which is below the percolation threshold of CNT/PU nanocomposites. The size of the GN layer affects the GFs of the flexible film sensors; a GN with a smaller size can achieve a greater GF. This study paves the way for the better application of different qualities of GN in flexible sensors.

**Keywords:** carbon nanotubes; graphene; film sensors; composite materials; polyurethane

## 1. Introduction

The emergence of film sensors has completely changed human life. The flexible sensors are continuously improved and innovated, which play a vital role in environmental monitoring [1], liquid detection [2], food detection [3], and other fields. The most important application is monitoring human health, collecting real-time characteristics of life parameters changes to understand human behavior better. Connecting the flexible sensor to the human skin can monitor pulse, heartbeat, blood pressure, finger bending, tiny facial muscle movements, vocal cord vibration, and so on [4–9], to complete multi-directional accurate monitoring of human life and health. In addition to applications in the medical and health field, flexible strain sensors can also be integrated into the intelligent robot arm, endowing the robotic arm with a human-like sense of touch, perceiving the size and spatial distribution of pressure [10].

Carbon materials have a broad prospect as conductive fillers for film sensors, especially graphene (GN) and carbon nanotubes (CNT), which have excellent electrical conductivity among carbon materials; these have attracted the extensive attention of researchers. The combination of GN or CNT with polymers in a specific way can form an electron transport channel in the polymer, while maintaining the flexibility of the polymer [11–15]. Vertuccio et al. [16] used simple film casting technology to produce CNT/epoxy resin composites for preparing strain sensors with GFs between 0.67 and 4.45. In order to increase the strain range of the sensor, using 3D printing technology, Christ et al. [17] prepared a multi-walled carbon nanotube/polyurethane highly elastic deformation sensor with a high GF.

Polyurethane (PU) elastomers have been proven to be used as substrates for film flexible sensors [18–21]. GN and CNT can be regularly and directionally arranged in

polyurethane through the CVD method [19], the biological stretching process [20], and 3D spraying [21]. The physicochemical properties of GN prepared by different methods are much different and they show different conductivities when doped into film sensors; however, there are few reports on the effects of GN prepared with different methods on the performance of film sensors.

In this paper, we prepared three kinds of graphene with different sheet sizes and thicknesses by the ultrasonic-assisted electrochemical method and chemical redox method. Different types of GN and CNT were dropped into PU to prepare PU-based composite films. The dispersion state of carbon nanomaterials in PU was characterized by SEM and XRD; the volume conductivity of PU composite flexible films was deeply analyzed; the GFs of PU flexible sensors were tested and calculated so that different qualities of GN can be better used in flexible sensors.

## 2. Materials and Methods

### 2.1. Materials

Graphite foils were provided by Jinglong Special Carbon Co., Ltd. (Beijing, China). NaOH and $(NH_4)_2SO_4$ were purchased from Tianjin Kemiou Chemical Reagent Co., Ltd. (Tianjin, China). Isophorone diisocyanate (IPDI) was purchased from Macklin Biochemical Technology Co., Ltd. (Shanghai, China). Polyether polyols were purchased from SINOPEC Tianjin Branch. Dibutyltin dilaurate (DBTDL) and L-ascorbic acid (L-AA, 99%) were purchased from Heowns Chemical Co., Ltd. (Tianjin, China). N, N-Dimethylformamide (DMF, AR, 99.5%) was provided by Bohua Chemical Reagent Co., Ltd. (Tianjin, China). Multi-walled carbon nanotubes (CNT, 99 wt%, OD = 5–15 nm, length = 10–30 um) were provided by Chengdu Organic Chemicals Co. Ltd (Chengdu, China).

### 2.2. Preparation of GNs

Three kinds of GN samples were prepared, one of which was prepared by the chemical redox method. Graphene oxide was prepared by the modified Hummers' process reported by the previous work of our team [22]; then, reduced graphene oxide (rGO) was obtained after being reduced with environmentally friendly L-AA at 25 °C for 24 h.

Another two GN samples were prepared by the ultrasonic-assisted electrochemical method in different electrolyte solutions (1.0 mol/L NaOH solution or 0.1 mol/L $(NH_4)_2SO_4$ solution). Graphite foils were used as the active electrodes; the insert electrode was platinum electrode (15 mm × 10 mm). The electrolysis time was 2 h. The black dispersion obtained by electrolysis was filtered under reduced pressure with a 0.22 μm polytetrafluoroethylene microporous filter membrane, and the solid was washed with distilled water until the pH of the filtrate was about 7. After drying, GN samples were obtained. The two GN samples prepared in 1.0 mol/L NaOH electrolyte solution and 0.1 mol/L $(NH_4)_2SO_4$ electrolyte solution were named G1 and G2, respectively.

### 2.3. Fabrication of PU-Based Films and Film Strain Sensors

A certain amount of GN and CNT powder were added into 100 mL of dewatered DMF, ultrasonically dispersed for 1 h. GN-CNT/DMF solution was mixed into polyurethane monomers with a certain ratio; then, the catalyst DBTDL was dripped. The polymerization was carried out at 85 °C for 3 h. To obtain GN-CNT/PU films, the GN-CNT/PU solutions were finally cured in PTFE mold at 60 °C for 4 h, and at 80 °C for 2 h to obtain films with a length of 4 cm, a width of 1 cm, and a thickness of 0.5–1.0 mm. Copper foil and silver wire were assembled on both ends of the film to obtain strain sensors. The film sample was named as Graphene$_{\text{volume fraction}}$-CNT$_{\text{volume fraction}}$/PU; for example, G1$_{0.01}$-CNT$_{0.2}$/PU means graphene G1-CNT/PU composite film, and the volume fraction of G1 and CNT are 0.01 vol% and 0.20 vol%, respectively. The schematic diagram of the preparation process of GN-CNT/PU film strain sensors is shown in Figure 1.

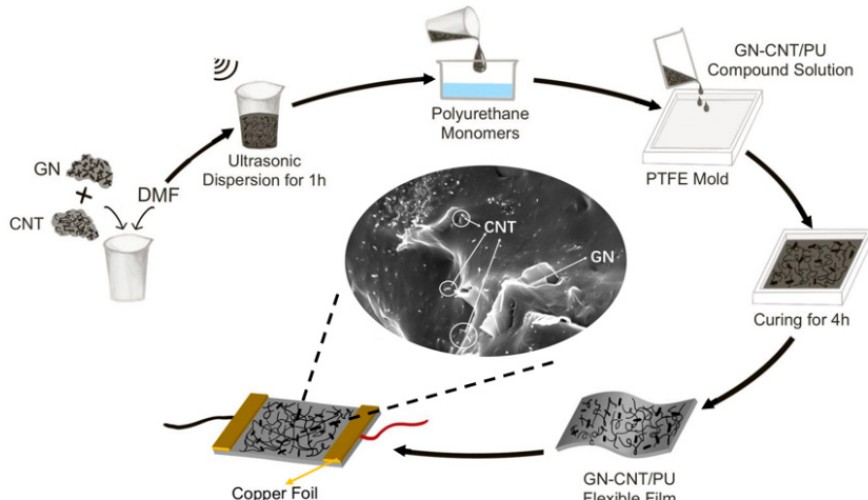

**Figure 1.** Schematic diagram of preparing GN-CNT/PU film strain sensors.

### 2.4. Characterizations

The surface morphology of graphene was observed by an atomic force microscope (AFM, Dimension Icon, Bruker, Karlsruhe, Germany) and field-emission scanning electron microscope (FESEM; Gemini 300, Carl Zeiss, Oberkochen, Germany). The chemical characteristics of prepared carbon materials was conducted by an X-ray photoelectron spectrometer (XPS; ESCALAB-250Xi, Thermo Fisher Scientific, Waltham, MA, USA). The dispersion of GN and CNT in PU films was characterized by XRD (MiniFlex600, Rigaku Corporation, Tokyo, Japan). The sensitivities of the film flexible strain sensors were tested by a universal testing machine (WDW-05L, Jinan Spai Technology Co., Ltd., Jinan, China), and the electrical conductivities were synchronously and automatically recorded by an electrochemical workstation (VERTEX V16407, Ivium Technologies, Eindhoven, The Netherlands) with a data acquisition frequency of 50 Hz.

## 3. Results and Discussion

### 3.1. Morphologies of GN

It can be seen from Figure 2a,d,g that the horizontal size of G1 is about 1 μm; the thickness is about 1.76 nm, around 3 to 4 layers of graphene; and the thickness of the graphene sheets is uniform, with the edges of G1 being relatively smooth. From Figure 2b,e,h, the horizontal size of G2 is about 0.7 μm and the sheets size is relatively uniform. The thickness of G2 is about 1.20 nm, with 2~3 layers of graphene. G2 has a smaller sheet size and fewer graphene layers than G1. The horizontal size of rGO sheets reduced with L-AA is about 1.5–2 μm and the thickness is about 2.00 nm in Figure 2c,f. The sizes of G1 and G2 are significantly smaller than that of rGO, and the defects of G1 and G2 are fewer; the surfaces are flatter and the structures are more complete.

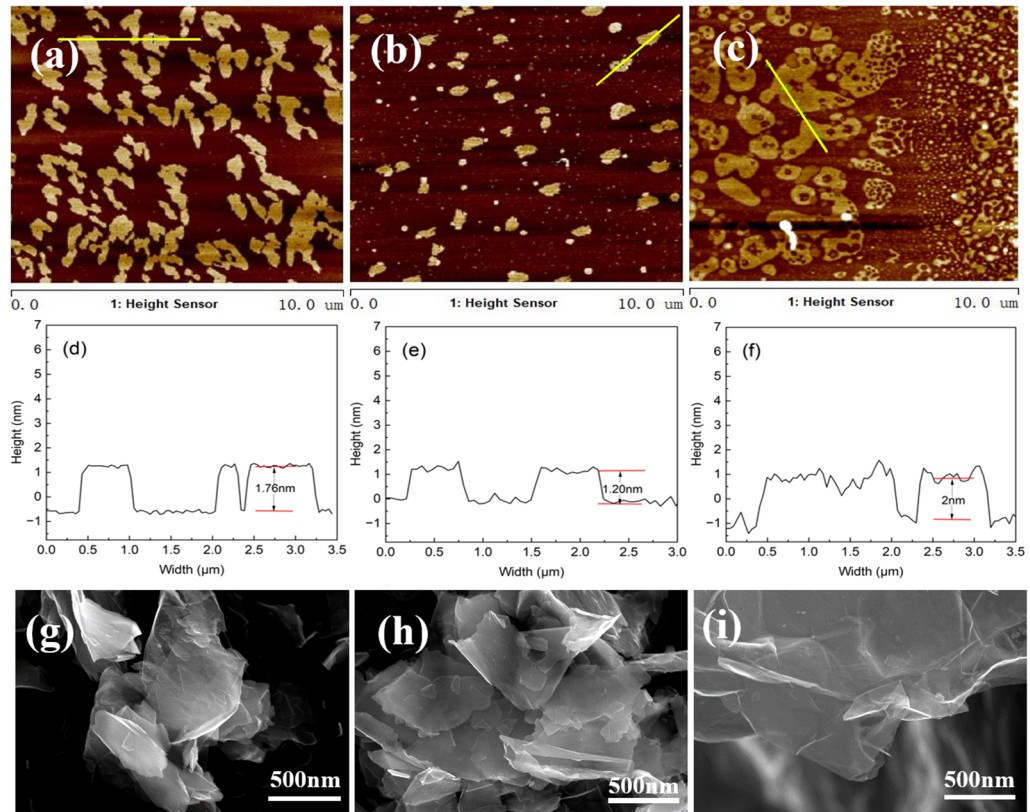

**Figure 2.** AFM 2D images of graphene (**a**) G1, (**b**) G2, and (**c**) rGO; the thickness (height) of graphene (**d**) G1, (**e**) G2, and (**f**) rGO; and FESEM images of graphene (**g**) G1, (**h**) G2, and (**i**) rGO.

### 3.2. Chemical Characteristics of Prepared GN and GO

The carbon and oxygen content in carbon material samples is usually obtained by XPS. By analyzing C 1s to O 1s peak in the XPS spectra of GO and GN, the atomic ratios of C/O were obtained. The precise oxygen content and C/O ratios of four carbon materials are listed in Table 1. It can be seen that the oxygen content of samples G1, G2, and rGO is increasing. The oxygen content of G1 is only 4.03 atom %, with a high C/O ratio of 23.81. The oxygen content of G2 (9.99 atom %) is slightly higher than that of G1. rGO reduced by L-AA contains more oxygen than GN prepared by the electrochemical method.

**Table 1.** XPS compositions of prepared GN and GO.

|  | C Content (Atomic %) | O Content (Atomic %) | C/O Ratios |
| --- | --- | --- | --- |
| G1 | 95.97 | 4.03 | 23.81 |
| G2 | 90.01 | 9.99 | 9.01 |
| GO | 68.04 | 31.96 | 2.12 |
| rGO | 89.44 | 10.56 | 8.47 |

Figure 3 shows the high-resolution spectra of C 1s of three kinds of graphene and GO. The peaks of C-C, C-OH, C=O, and (C=O)-O groups are about 284.6 eV, 285.5 eV, 287.6 eV, and 288–290 eV, respectively [23–25]. Figure 3c shows that the sample GO contains a large number of oxygen-containing functional groups, with the oxygen content of 31.96%. The peak intensity of oxygen-containing functional groups in rGO is much weaker than that of GO, indicating that some oxygen-containing functional groups in GO have been successfully reduced by L-AA.

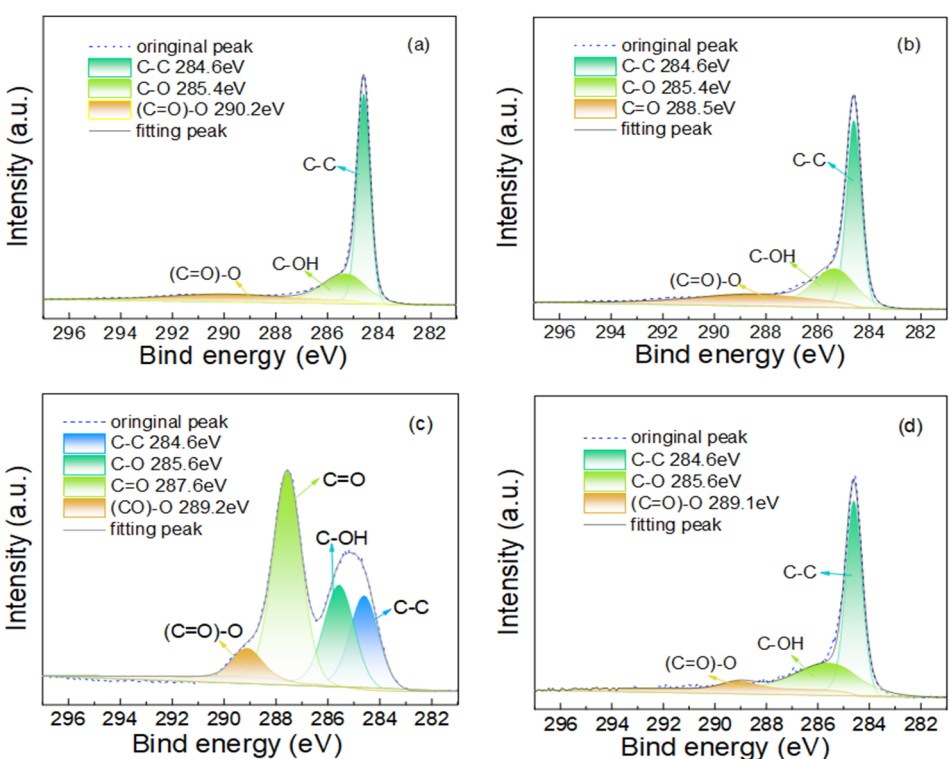

**Figure 3.** Curve-fitting of C 1s XPS spectra of (**a**) G1, (**b**) G2, (**c**) GO, and (**d**) rGO.

### 3.3. Dispersion of GN and CNT in PU Films

Figure 4a–d are FESEM images of the cross section in thickness direction of PU and $CNT_{0.2}$/PU films, respectively. The cross section of the PU film is relatively clean; only brittle fracture lines can be observed. The cross section of the sample $CNT_{0.2}$/PU exposes a lot of carbon nanotubes (bright spots and lines), indicating that the carbon nanotubes can be uniformly mixed in the PU through ultrasonic and mechanical stirring. From Figure 4c–j, it can be observed that many carbon nanotubes are randomly dispersed around GN sheets with a light yarn-like structure, which is the connection between the sheet-shaped GN and the linear carbon nanotubes to form a conductive path. The addition of graphene prevents CNT from agglomeration in PU to varying degrees. By comparing with Figure 4e,i,j the agglomeration of CNT in G2-doped composite film is significantly less than that in G1- and rGO-doped films.

The dispersion degree of graphene and carbon nanotubes in PU films can be explained laterally by XRD patterns in Figure 5. As expected, it is observed that the carbon nanotubes exhibit broad diffraction peaks at 25.8° and 43.1°; the former is the diffraction peak of the (002) crystal plane perpendicular to the diameter of the carbon nanotube, and the latter is the characteristic diffraction peak of the (100) crystal plane parallel to the diameter of the carbon nanotube. Graphene G1 and G2 have obvious diffraction peaks at 2θ = 26.5°, and rGO has characteristic peaks at 2θ = 23.6°. The diffraction peak of pure PU film appears near 2θ = 19.1°, which is mainly related to the short-range orderly and regular structure of the hard and soft segments of PU, and the existence of the disordered structure of the PU amorphous phase. After CNT and GN are doped in PU film, the XRD patterns of the composites show the characteristic peak of PU at 19.1°, while the characteristic peaks of CNT and GN disappear. It can be explained that when GN and CNT are doped in PU, the stacked GN sheets and agglomerated CNT will disperse under the action of Van der Waals force. Therefore, the crystal lattice of GN and CNT will be distorted, the XRD peaks will broaden, and the peaks intensity will decrease. When crystallite size become smaller with the increase of crystal lattice distortion, carbon material will be an amorphous structure, completely disordered, and the diffraction peaks of GN and CNT disappear. Therefore,

it can be said that the graphene and carbon nanotubes can be well dispersed in PU film through ultrasonic dispersion and mechanical dispersion.

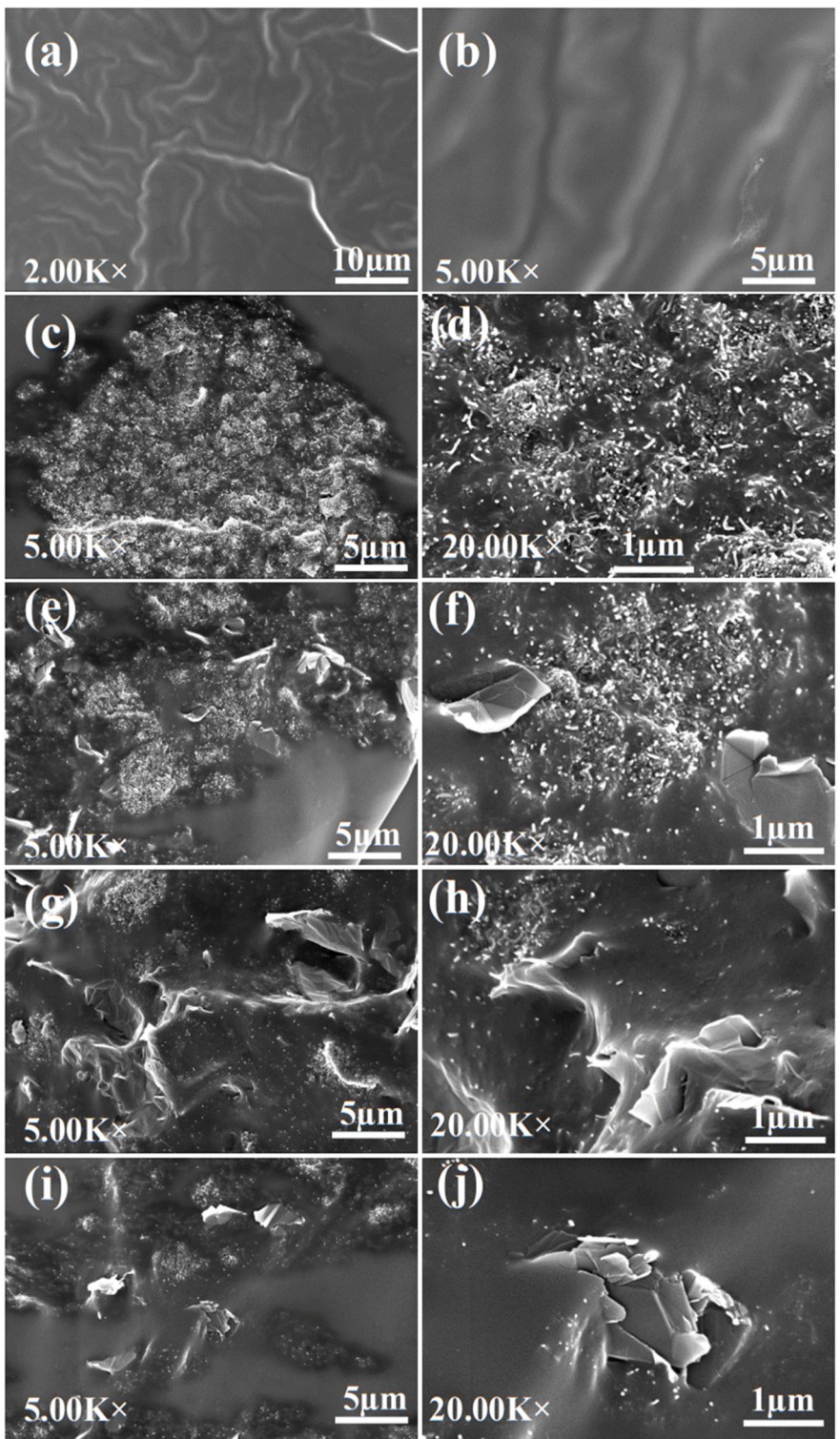

**Figure 4.** FESEM images of the cross section in thickness direction of PU and modified PU conductive films with different GN and CNT: (**a,b**) PU; (**c,d**) $CNT_{0.2}$/PU; (**e,f**) $G1_{0.1}$-$CNT_{0.2}$/PU; (**g,h**) $G2_{0.1}$-$CNT_{0.2}$/PU; (**i,j**) $rGO_{0.1}$-$CNT_{0.2}$/PU.

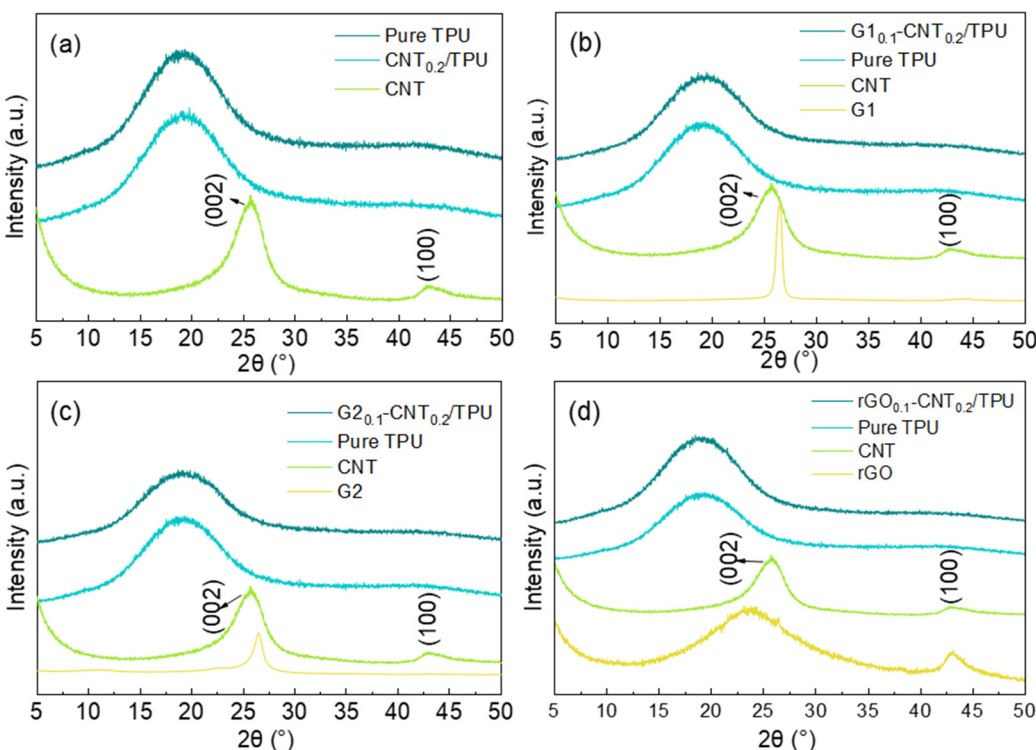

**Figure 5.** XRD patterns of pure PU film and PU composite films, with GN and CNT (**a**) $CNT_{0.2}/PU$, (**b**) $G1_{0.1}$- $CNT_{0.2}/PU$, (**c**) $G2_{0.1}$-$CNT_{0.2}/PU$, and (**d**) $rGO_{0.1}$-$CNT_{0.2}/PU$.

*3.4. Electrical Properties of GN-CNT/PU Composite Films and Flexible Sensors*

According to the percolation theory, the electrical conductivity of the composite and the volume fraction of the filler usually satisfy the following equation:

$$\sigma = \sigma_0(V - V_c)^t$$

where σ and $\sigma_0$ are the volume conductivities of composites and the fillers (S/cm); *V* and $V_c$ are the volume fraction and volume percolation threshold of the fillers (%); and *t* is a constant, relating to the dimension of conductivity path.

It can be seen from Figure 6a that when the content of CNT increased from 0.127 vol% to 0.255 vol%, the volume conductivities of the composites increased from $5.13 \times 10^{-12}$ S/cm to $5.12 \times 10^{-7}$ S/cm, having a 5-order magnitude improvement. According to the percolation theory, the volume percolation threshold of the fillers ($V_c$, %) of CNT doped in PU film is about 0.296 vol%. The constant *t* of CNT/PU flexible films is 1.16, calculated by percolation theory from Figure 6b. It can be said that CNT forms a good 2D conductive network in PU substrate and has a low percolation threshold.

The experiments set the doping content of CNT as 0.200 vol%, far less than that of its percolation threshold. The volume conductivity of the sample $CNT_{0.2}/PU$ film is $2.19 \times 10^{-11}$ S/cm. As shown in Figure 7a, the volume conductivity of the GN-$CNT_{0.2}/PU$ film increased with the addition of GN. When the addition amount of G1, G2, and rGO was 0.020 vol% (1/10 of CNT content), the volume conductivities of $GN_{0.02}$-$CNT_{0.2}/PU$ films were $5.77 \times 10^{-7}$ S/cm, $1.41 \times 10^{-7}$ S/cm, and $1.70 \times 10^{-6}$ S/cm, for G1, G2, and rGo, respectively. Compared with $CNT_{0.2}/PU$, doping a minute amount of GN in $CNT_{0.2}/PU$ can make the volume conductivities of flexible films increased by 4 to 5 orders of magnitude, and the effect of enhancing the conductivity is obvious. With the further increase of the addition of GN, the conductivities of composites increase linearly. It can be seen from Figure 7b that the constant t (slopes of the lines in Figure 7b) of G1, G2, and rGO in the $CNT_{0.2}/PU$ system are 2.06, 1.31, and 1.36, respectively, which are higher than the t value of the CNT/PU film. It can confirm the role of graphene in CNT/PU composite film, as

expected, forming a stronger 2D and 3D hybrid conductive network, or 3D conductive network with CNT. Compared with the need to dope about 0.100 vol% CNT to obtain a decent volume conductivity, adding less than 0.020 vol% of graphene can obtain the same conductive effect. This phenomenon may be caused by the smaller graphene flakes; this not only can assist in forming a more complete conductive path, but also prevent CNT entanglement or formation of CNT bundles, making the dispersion of CNT more uniform.

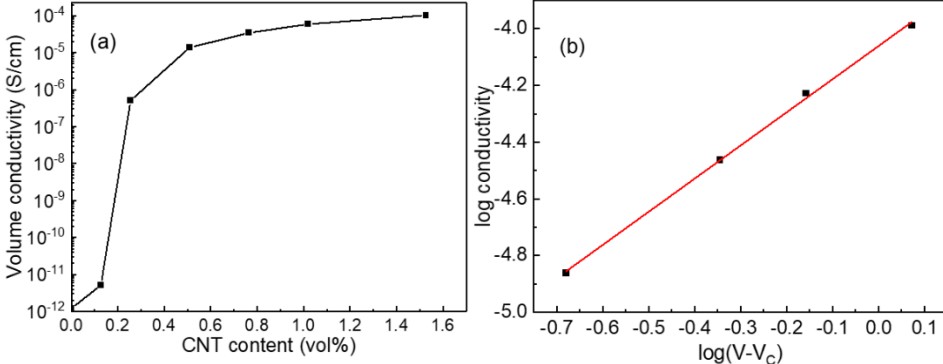

**Figure 6.** (**a**) Volume conductivity plotted as a function of CNT content; (**b**) log($\sigma$) plotted as a function of log(V-Vc).

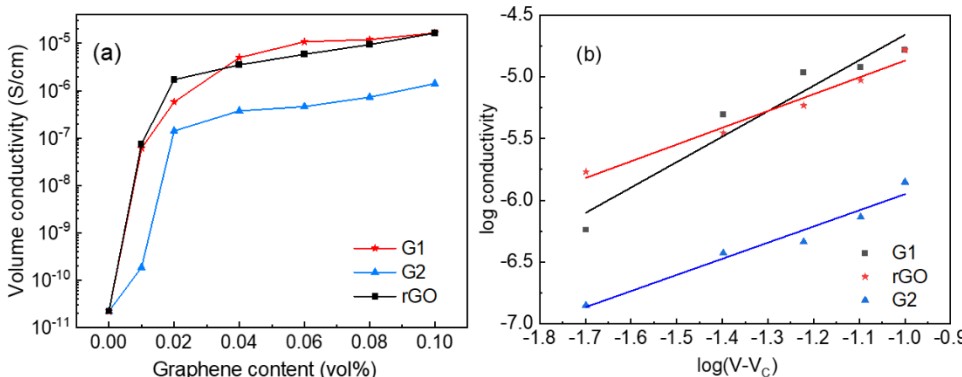

**Figure 7.** (**a**) Volume conductivity of GN-CNT$_{0.2}$/PU composite film as a function of the volume fraction of GN; (**b**) log($\sigma$)-log(V-V$_C$) curves of CNT$_{0.2}$/PU composite film doped with different kinds of graphene.

From Figure 8a, the GFs of G1$_x$-CNT$_{0.2}$/PU sensors (x = 0.01, 0.02, 0.04, 0.06) are 1.14, 4.18, 13.15, and 1.88, respectively, showing a trend of increasing firstly and then, decreasing. To explore the influence of graphene prepared by different methods on the GFs of the flexible sensors, the current change of the GN$_{0.06}$-CNT$_{0.2}$/PU sensors were tested within a strain of 20%, and the results are shown in Figure 8b. The GFs of G1$_{0.06}$-CNT$_{0.2}$/PU, G2$_{0.06}$-CNT$_{0.2}$/PU, and rGO$_{0.06}$-CNT$_{0.2}$/PU are 1.88, 3.47, and 1.41, respectively. The G2-doped film flexible sensor has the highest GF. From the AFM results of GN, G2 has the smallest sheet size, followed by G1, and rGO has the largest size. It is assumed that CNTs in PU are untwisted, oriented, and without overlapping distribution. In an ideal state, according to the outside diameter and length of carbon nanotubes, when the volume fraction of CNT in PU film is 0.2 vol%, the estimated gap between carbon nanotubes is about 200–500 nm. The smaller the graphene sheet, the greater the probability of graphene being inserted into the gap between the CNT. The greater the overlap density with the CNT, the more obvious the resistance change during stretching; and therefore, the greater the GF. When the GN layer is large, according to estimated gap between CNT, the smaller the probability of GN inserting into the gap, the less the hybrid conductive network formed between GN and CNT; thus, GF will also be smaller.

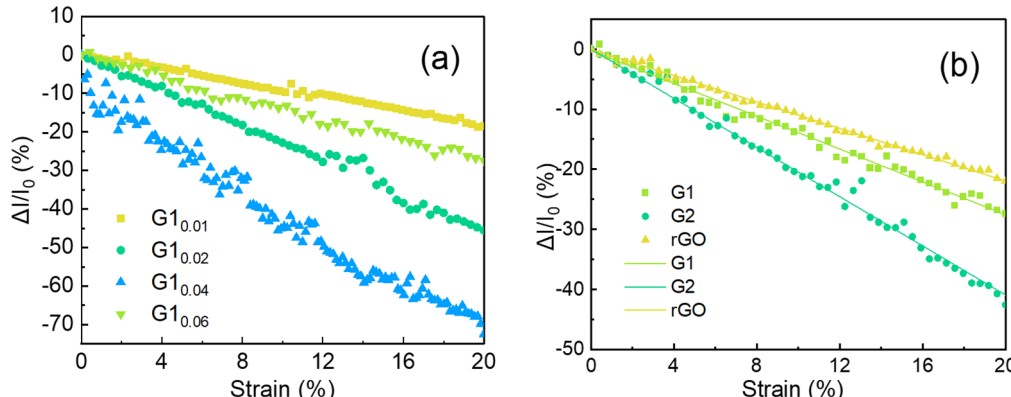

**Figure 8.** (**a**) Normalized current change $\Delta I/I_0$ plotted as a function of tensile strain (20%) of $G1_x$-$CNT_{0.2}$/PU composites; (**b**) $\Delta I/I_0$ plotted as a function of tensile strain (20%) of $GN_{0.06}$-$CNT_{0.2}$/PU composites.

The performance comparison between the flexible sensors assembled in this work and the PU-based sensors previously reported are shown in Table 2. It is found that, compared to others' works that use PU as substrate, and the combination of conductive fillers and PU in a simple and uniform mixture, the flexible sensors of this work can reach the GF at a higher level when the content of the added conductive fillers is very small.

**Table 2.** Performance of flexible sensors based on PU matrix.

| Conductive Fills | Content | Strain (%) | GF | References |
|---|---|---|---|---|
| Carbon black | 3 wt% | 30% | 3.5 | [26] |
| GN | 0.6 wt% | 30% | 0.78 | [13] |
| CNT | 2 wt% | 5% | 20 | [27] |
| rGO/$Fe_3O_4$ | 1 wt% | 12% | 6.2 | [18] |
| G1/CNT | 0.04 vol%/0.2 vol% | 20% | 13.15 | This work |

Cyclic tensile tests (strain from 15% to 50% and circulated 100 times) were carried out on the $G1_{0.06}$-$CNT_{0.2}$/PU sensor. It can be seen from the Figure 9a that after the sample is stretched for dozens of times, the curve tends to be flat; however, there are still fluctuations. $\Delta R/R_0$-time and the corresponding elongation–time curves of the sample after 21–25 cycles and 91–95 cycles are shown in Figure 9b. It can be observed that the normalized resistance of the sample increases with the elongation of the $G1_{0.06}$-$CNT_{0.2}$/PU film sensor. When the sensor elongation is 50%, the change in resistance value reaches the maximum; that is, a higher peak in the curve. As the elongation of the sample decreases, the value of $\Delta R/R_0$ decreases first and then, increases; in addition, a secondary peak appears when the strain recovers to 15%. The sensors made of G2- and rGO-doped CNT/PU films showed almost the same secondary peak phenomenon in the cycle tests; however, the G1-CNT/PU film sensor showed better stability during the long cyclic tensile tests. Therefore, the cyclic test results of the G1-CNT/PU film sensor were reported. The main reason for the secondary peak is that during the rebound process of the composite, although the conductive network reconstruction in the sample mainly occurs as the strain decreases, the conductive network of the carbon material will be subjected to secondary damage due to residual stress and creep. When the residual stress is greater than the reconstruction of the conductive network in the PU itself, the resistance will increase again, and the secondary peak appears.

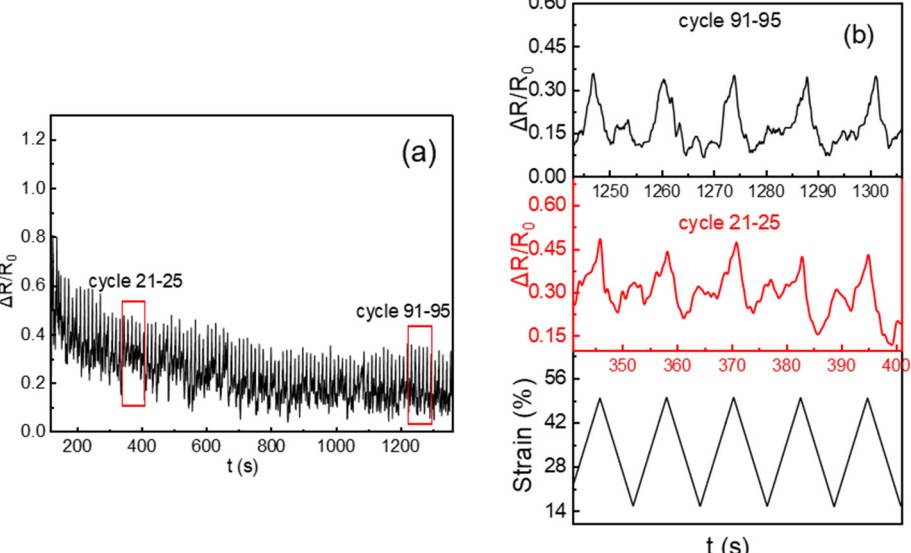

**Figure 9.** (**a**) normalized resistance change $\Delta R/R_0$-time curves in correspondence of strain varies from 15% to 50% of $G1_{0.06}$-$CNT_{0.2}$/PU composite; (**b**) $\Delta R/R_0$-time curves and tensile strain–time curve of $G1_{0.06}$-$CNT_{0.2}$/PU sensor.

## 4. Conclusions

Three kinds of GN samples with different sheet sizes and thicknesses were prepared and composited with CNT and PU to manufacture film flexible sensors. The size of the GN layers affected the GF of the flexible sensors; that means the appropriate size of a GN sheet is essential to obtain a flexible sensor with an excellent GF. With the same amount of graphene added, a higher GF of sensors can be obtained by reducing graphene size. These results have an excellent reference for the application of graphene with different sheet sizes in flexible sensors.

**Author Contributions:** Conceptualization, C.L. and C.Y.; data curation, C.L.; formal analysis, C.L.; investigation, C.L., J.G. and Y.M.; methodology, C.L., X.G. and C.Y.; project administration, C.Y.; resources, X.G. and C.Y.; supervision, C.Y.; validation, C.L., J.G. and Y.M.; visualization, C.L.; writing—original draft, C.L.; writing—review and editing, X.G. and C.Y. All authors have read and agreed to the published version of the manuscript.

**Funding:** This research was funded by the Independent Innovation Fund of Tianjin University (No. 2021XZS-0017).

**Institutional Review Board Statement:** Not applicable.

**Informed Consent Statement:** Not applicable.

**Data Availability Statement:** Not applicable.

**Conflicts of Interest:** The authors declare no conflict of interest.

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
