# Peer review of "Effects of Graphene Morphology on Properties of Carbon Nanotube/Polyurethane Film Strain Sensors"

_coatings, doi:10.3390/coatings12121889_

Round 1
Reviewer 1 Report
1. Please read carefully line no. 91
2. Mention the characterization name and adequately write all structure dimensions in figure2. Please write all the paragraphs in the morphology part correctly.
3. It is not an SEM, called FESEM line no. 116, 147, 151).
4. Write all structure dimensions in TEXT and Figure 3.
5. Please mention the plane of rGO and PU in the text.
6. Mention the plane of all phases in Figure 4 also.
7. Provide HRTEM and EDX to confirm all phases and composition.
8. Some of the sentences are unclear, e.g., In conclusion: When GN with the same volume fraction is doped, using GN with a smaller size can obtain surprising GF.
9. please explain the terms: high gauge factor, strain range, percolation threshold, etc.
10. The title should mention that it is a strain sensor.
11. I am suggesting a few good articles to refer, which can help you in better explaining the results (citation is not necessary, unless essential)
Improved Nanoindentation and Microwave Shielding Properties of Modified MWCNT Reinforced Polyurethane Composites
TK Gupta,
J. Mater. Chem. A 1 (32), 9138-9149
Superior nano-mechanical properties of reduced graphene oxide reinforced polyurethane composites
TK Gupta,
RSC Advances 5, 16921-16930
Exploring the possibility of using MWCNTs sheets as an electrode for flexible room temperature NO 2 detection
R Kumar, Mamta,
Superlattices and Microstructures 164, 107165
Faster response of NO2 sensing in graphene–WO3 nanocomposites
S Srivastava, K Jain, VN Singh, S Singh, N Vijayan, N Dilawar, G Gupta, ...
Nanotechnology 23 (20), 205501
Enhanced electrochemical biosensing efficiency of silica particles supported partially reduced graphene oxide for sensitive detection of cholesterol
S Abraham,
Journal of Electroanalytical Chemistry 757, 65–72
Nitrogen doped high quality CVD grown graphene for fast responding NO2 gas sensor
S Srivastava
New journal of Chemistry 42, 9550-9556
Partially reduced graphene oxide-gold nanorods composite based bioelectrode of improved sensing performance
NR Nirala,
Talanta 144, 745-754
Author Response
Comment 1: Please read carefully line no. 91.
Response: Thanks for your valuable feedback. We have added the description for the sample names in lines 92-95.
Comment 2: Mention the characterization name and adequately write all structure dimensions in figure2. Please write all the paragraphs in the morphology part correctly.
Response: We have checked all the morphology data of GN. In order to observe the morphology of GN more intuitively and clearly, we have made appropriate changes to the thickness graph of GN sheet in Figure 2.
Comment 3: It is not an SEM, called FESEM line no. 116, 147, 151).
Response: Thank you for reminding, we have made corrections.
Comment 4: Write all structure dimensions in TEXT and Figure 3.
Response: We have made the correction in text and Figure 4 (orginal Figure 3).
Comment 5: Please mention the plane of rGO and PU in the text.
Response: From the FESEM images of GN, it can be observed that graphene sheets are curled and cross agglomerated and little ordered stacking can be seen. In the XRD patterns of GN, only a small broad diffraction peak is displayed. Besides, the residual oxygen-containing functional groups, lamellar structural defects and disordered stacking of graphite sheets in GN samples cause the characteristic diffraction peak (002) of graphite to shift to the left, and new diffraction peaks of GN are formed near 2θ= 23-26°. The diffraction peak of pure PU film is mainly related to the short-range orderly and regular structure of the hard and soft segments of PU, no specific lattice is formed in the test environment.
Comment 6: Mention the plane of all phases in Figure 4 also.
Response: We have marked the crystal plane in XRD patterns.
Comment 7: Provide HRTEM and EDX to confirm all phases and composition.
Response: We agree with the reviewer that HRTEM and EDX would be helpful to confirm all phases and composition. We believe that if there is a better scientific research environment, we will try our best to do these tests. However, due to the impact of epidemic control in China recently, we may not have available experimental equipment to do these additional tests. In fact, in order to observe the interaction between GN and CNT in PU and analyze the composition of GN, we have conducted SEM, XRD and XPS analysis on the samples. These test results also have certain reference significance. we will do more in-depth tests and research in these direction in the future.
Comment 8: Some of the sentences are unclear, e.g., In conclusion: When GN with the same volume fraction is doped, using GN with a smaller size can obtain surprising GF.
Response: Maybe the word "surprising" is ambiguous and confused, and we directly changed it to "higher".
Comment 9: Please explain the terms: high gauge factor, strain range, percolation threshold, etc.
Response:
- High gauge factor: The change of force in the film sensor leads to deformation in the film, thereby causing its electrical resistance to change. The fractional change in resistance, ∆R/R0, is related to the mechanical strain ε by the gauge factor (GF):
GF=(∆R/R0)/ε
When the substrate of the sensor is PU and the conductive fillers are carbon materials, the flexible sensors of this work can reach the GF a higher level when the content of conductive fillers added is very small. We have added the part comparing with other research results in line Table 2.
- Strain range: GF obtained without a strain range is meaningless. When the polymer substrate of the flexible sensor has a large elastic deformation, a higher GF is often obtained. When calculating GF, the strain range is generally limited within the recoverable deformation range of the strain sensor.
- Percolation threshold: The percolation threshold is defined as the value of the solid content above which the rheological properties increase in an exponential way. According to the percolation theory, the curves of the conductivity of the composite and the volume fraction of the filler can be drawn. With the increase of the amount of conductive filler, the conductivity of the composite will suddenly increase by several orders of magnitude. The filling amount of conductive filler corresponding to the conductivity mutation point (the line with the largest conductivity change and the line with the conductivity tending to be stable are used as extension lines, and the intersection point of the extension lines is the conductivity mutation point) of the complex is the percolation threshold.
Comment 10: The title should mention that it is a strain sensor.
Response: It is a very good suggestion. We have revised the title of the paper.
Comment 11: I am suggesting a few good articles to refer, which can help you in better explaining the results (citation is not necessary, unless essential).
Response: We thank you for the references you provided. We have learned a lot from the more research directions of sensors and from the analysis methods used in all the literatures, and have cited some of them in Results and Discussion section.
Reviewer 2 Report
The paper reports the fabrication of flexible strain sensors based on PU/ CNT composites with different types of GN. The authors show an important effect of GN in reduction of the percolation threshold of CNT/PU nanocomposites. The findings are interesting, and the manuscript is well organized, but in my opinion some points should be addressed before the publication. Below I give some suggestions to improve the manuscript.
- The methods are not well described, and some information is missing regarding the characterization techniques and the fabrication. For example: which is the dimension and thickness of the prepared films?
- There is a lack of characterization of different GN sheets, except for morphology. A chemical characterization of GN should be added to confirm the effective reduction of GO and to determine the presence of surface functionalities, that can affect the dispersion of the fillers inside the matrix. The authors should consider to add XPS or Raman measurements, for example. In addition, the discussion about the XRD pattern of the G1, G2 and rGO could be improved.
- The quality of some images should be improved. It is difficult to read the dimension in the AFM profiles in Figure 2 a-b-c.
- Figure 3. According to the caption, the samples in figure b-c-d-e have the same theoretical amount of CNT (0.2%), corresponded to the bright spots. However, there are differences between the samples. Can the authors justify these differences? From my perspective, these differences could be an indication of agglomeration. I would suggest to add some images at lower magnification to have a more comprehensive representation of the samples. In addition, the unit in the scale bar should be changed in “µm”.
- Figure 7a: the film shows a trend of increasing first and then decreasing with G1 amount. How the authors explain this trend?
- Figure 7c: the scales in the inset are difficult to read. Is the up curve corresponding to the 91-95 cycles? It seems more regular compared to the below one (21-25 cycles). How do the authors explain this behaviour? And why the authors didn’t report cyclic measurements for samples with G2 and rGO? Do they show a similar double peak?
- The results of sensing measurements should be compared with similar systems (i.e. polymer systems filled with GN or CNT) found in the literature, so that the readers could get some idea how important the improvement really is.
Author Response
Comment 1: The methods are not well described, and some information is missing regarding the characterization techniques and the fabrication. For example: which is the dimension and thickness of the prepared films?
Response: Thank you for your valuable comment, which can help us improve the paper so that readers can better understand the Fabrication of PU-based films and thin film strain sensors. We have described the preparion of the thin film strain sensors in more detail in line 90-96.
Comment 2: There is a lack of characterization of different GN sheets, except for morphology. A chemical characterization of GN should be added to confirm the effective reduction of GO and to determine the presence of surface functionalities, that can affect the dispersion of the fillers inside the matrix. The authors should consider to add XPS or Raman measurements, for example. In addition, the discussion about the XRD pattern of the G1, G2 and rGO could be improved.
Response: We have supplemented the section 3.2: Chemical characteristics of prepared GN and GO. The sample element content was tested and the XPS spectra were analyzed in this part.
Comment 3: The quality of some images should be improved. It is difficult to read the dimension in the AFM profiles in Figure 2 a-b-c.
Response: We have changed the layout of the graph to make the GN dimension clearer and more readable in Figure 2.
Comment 4: Figure 3. According to the caption, the samples in figure b-c-d-e have the same theoretical amount of CNT (0.2%), corresponded to the bright spots. However, there are differences between the samples. Can the authors justify these differences? From my perspective, these differences could be an indication of agglomeration. I would suggest to add some images at lower magnification to have a more comprehensive representation of the samples. In addition, the unit in the scale bar should be changed in “µm”.
Response: Your advice of Figure 3 gives us a more comprehensive understanding of the interaction between graphene and carbon nanotubes in PU substrate. As you suggested, we added some low-resolution SEM images. From the Figure 4, we can clearly observe agglomeration degree of carbon nanotubes in GN-CNT/PU composite films, and this phenomenon is consistent with our calculation results in line 220-224. Thank you again for pointing out this, which makes the manuscript’s logic more self-consistent.
Comment 5: Figure 7a: the film shows a trend of increasing first and then decreasing with G1 amount. How the authors explain this trend?
Response: When the amount of G1 added is small, only a little number of conductive paths can be formed in the composites, there is a large distance between the graphene and the carbon nanotubes, so the GF is low. As the amount or the concentration of G1 increases, the spatial distance between the graphene sheets and carbon nanotubes decreases, and the probability of overlapping increases, forming more conductive paths. Under the action of external stress, a larger resistance change can be produced, so the GF of the sensor increases. When the content of G1 is increased to a certain volume fraction, the GF of the sample is not increased as expected, mainly due to the high density of the conductive pathways of graphene and carbon nanotubes in the film, and a small strain cannot make the conductive pathways form more breaking point.
Comment 6: Figure 7c: the scales in the inset are difficult to read.
- Is the up curve corresponding to the 91-95 cycles?
- It seems more regular compared to the below one (21-25 cycles). How do the authors explain this behaviour?
- And why the authors didn’t report cyclic measurements for samples with G2 and rGO? Do they show a similar double peak?
Response:
We have enlarged Figure 7c to Figure 9.
At the beginning of the strain sensor under stress, the sensor will have a stable period of output current, which may be related to the construction and reorganization of the conductive network. When a stable conductive network is formed inside the sensor, a stable electrical signal can be output with the change of stress.
The sensors made of three kinds of graphene showed almost the same double peak phenomenon in the cycle test, but G1 showed better stability during the long Cyclic tensile test. Therefore, the cyclic test results of G1-CNT/PU sensor were reported in our MS.
Comment 7: The results of sensing measurements should be compared with similar systems (i.e. polymer systems filled with GN or CNT) found in the literature, so that the readers could get some idea how important the improvement really is.
Response: We appreciate your suggestions. We have added Table 2 and related text in revised manuscript.
Round 2
Reviewer 1 Report
Since the XPS of carbon is done, what is the reference for calculating the peak position in the XPS experiment?
In XRD, peak broadening has some meaning. Please explain: the spreads of the XRD peaks
Author Response
Comment 1: Since the XPS of carbon is done, what is the reference for calculating the peak position in the XPS experiment?
Response: We have added two references to XPS spectra analysis in line 136.
Comment 2: In XRD, peak broadening has some meaning. Please explain: the spreads of the XRD peaks
Response: The possible reasons of XRD peak broadening are instrument broadening, crystallite size and micro-strain. We think that the XRD peak broadening of GN and CNT are mainly caused by crystallite size and micro-strain. “When GN and CNT are doped in PU, the stacked GN sheets and agglomerated CNT will disperse under the action of Van der Waals force. Therefore, the crystal lattice of GN and CNT will be distorted, the XRD peaks will broaden and the peaks intensity will decrease. When crystallite size become smaller with the increase of crystal lattice distortion, the carbon material will be amorphous structure, completely disordered, and the diffraction peaks disappear.” The above explanation has been added in line 165-171.
Reviewer 2 Report
I will suggest removing the term "thin" from the description and leaving a general "film" since the thickness is quite high (0.5 - 1 mm) to be considered a thin film (usually in the range of a few nanometers to several micrometres).
My second suggestion is to add a sentence that summarises the choose to report only the cyclic test for G1 in the text (see your response to comment 6).
Author Response
Comment 1: I will suggest removing the term "thin" from the description and leaving a general "film" since the thickness is quite high (0.5 - 1 mm) to be considered a thin film (usually in the range of a few nanometers to several micrometres).
Response: Thank you for your comments and we have made appropriate revisions to the full text.
Comment 2: My second suggestion is to add a sentence that summarises the choose to report only the cyclic test for G1 in the text (see your response to comment 6).
Response: We have supplemented the selection of sample in cyclic test in line 256-259.